# COVID-19: Short term prediction model using daily incidence data

**Hongwei Zhao** [1]*, **Naveed N. Merchant**[2], **Alyssa McNulty**[1], **Tiffany A. Radcliff**[1], **Murray J. Cote**[1], **Rebecca S. B. Fischer**[1], **Huiyan Sang**[2], **Marcia G. Ory**[1]

**1** School of Public Health, Texas A&M University, College Station, TX, United Stated of America,
**2** Department of Statistics, Texas A&M University, College Station, TX, United Stated of America

* hongweizhao@tamu.edu

## Abstract

### Background

Prediction of the dynamics of new SARS-CoV-2 infections during the current COVID-19 pandemic is critical for public health planning of efficient health care allocation and monitoring the effects of policy interventions. We describe a new approach that forecasts the number of incident cases in the near future given past occurrences using only a small number of assumptions.

### Methods

Our approach to forecasting future COVID-19 cases involves 1) modeling the observed incidence cases using a Poisson distribution for the daily incidence number, and a gamma distribution for the series interval; 2) estimating the effective reproduction number assuming its value stays constant during a short time interval; and 3) drawing future incidence cases from their posterior distributions, assuming that the current transmission rate will stay the same, or change by a certain degree.

### Results

We apply our method to predicting the number of new COVID-19 cases in a single state in the U.S. and for a subset of counties within the state to demonstrate the utility of this method at varying scales of prediction. Our method produces reasonably accurate results when the effective reproduction number is distributed similarly in the future as in the past. Large deviations from the predicted results can imply that a change in policy or some other factors have occurred that have dramatically altered the disease transmission over time.

### Conclusion

We presented a modelling approach that we believe can be easily adopted by others, and immediately useful for local or state planning.

**Data Availability Statement:** We used data from public available source, namely, COVID-19 Data Repository by the Center for Systems Science and Engineering (CSSE) at Johns Hopkins University https://github.com/CSSEGISandData/COVID-19.

**Funding:** The author(s) received no specific funding for this work.

**Competing interests:** The authors have declared that no competing interests exist.

## Introduction

Since the World Health Organization declared a pandemic for the novel SARS-CoV-2 2019 virus (COVID-19) on March 11, 2020 [1], the Americas, Europe, South-East Asia and Eastern Mediterranean regions have the most documented cases [2]. Globally, nationally, and at every sub-governmental level, there is a need to monitor the current caseload and project the rate and nature of the spread to guide public health awareness, preparedness, and response. Societies have to deal with many pressing issues such as ensuring adequate supplies of personal protective equipment, considerations about the adequacy of the health care workforce and other health care resources, as well as how to balance restrictive safety guidelines with keeping businesses open and the economy sound. For a novel infectious disease, it is especially important to forecast future cases based on what has happened in the immediate past.

Prediction for the number of cases in a pandemic and implications for health care needs and resources have received a lot of attention in the scientific world [3–5], government agencies [6–8], and in media lately [9–11]. With the plethora of models, there is also growing scrutiny [12] about the accuracy of different models, and an appreciation that model parameters need to be refined based on evolving knowledge about the disease trajectory and factors impacting infection and transmission rates.

The different approaches to modeling and forecasting infectious disease epidemics can be characterized as: 1) mechanistic models based on SEIR (referring to Susceptible, Exposed, Infected, and Recovered states) framework [13]; or its modified version [14–16]; 2) time series prediction models such as ARIMA [17], Grey Model [18], and Markov Chain models [19]; and 3) agent type models (i.e. simulating individual activities for a population) [20]. Even within each category, there are different types of approaches attempted. For SEIR models, there are deterministic models involving differential equations, and stochastic models entailing probability distributions. There are models that are designed to make long-term forecasts, and models that are best used for short-term predictions. For this paper, we primarily focus on short-term predictions based on SEIR concepts intended to forecast incidence cases for the next two to three weeks.

The SEIR model is an extension of the classical SIR model [21], and both SEIR and SIR models are foundations for many epidemiological modeling techniques. The model's strength lies in its simple approximation of a complex process. For example, a typical SIR model specifies that at a certain time $t$, the population (with size $N$) can be classified as people who are susceptible $S(t)$, infected $I(t)$, and recovered $R(t)$ according to the following series of differential equations:

$$\frac{dS(t)}{dt} = -\beta(t)I(t)\frac{S(t)}{N},$$

$$\frac{dI(t)}{dt} = \beta(t)I(t)\frac{S(t)}{N} - \lambda(t)I(t),$$

$$\frac{dR(t)}{dt} = \lambda(t)I(t),$$

$$S(t) + I(t) + R(t) = N,$$

where $\beta$ and $\lambda$ represent the transmission rate and recovery rate, respectively.

In theory, the population size for each state as a time series can be used to estimate the parameters in the model according to the system of equations. In practice, modelers rarely

have an accurate count of people at each stage, and the parameters could change with time. The problem has been tackled using different approaches. For example, Zhu and Chen [22] considered a statistical transmission model for early phase of COVID-19 outbreak; Wu et al. [23] incorporated the possibility of people moving out of the compartments due to migration in the modified SEIR model. However, both approaches made the assumption that the transmission rate was constant. In many states within US, or in many counties, we have seen a rapid change of the transmission rate caused by public behavior and public policy, therefore, it is not realistic to use a model with a constant transmission rate over a long period of time.

Although many approaches to predicting infectious disease transmission have appeared in literature, we have not found one method that can be used readily for a day-to-day short-term forecast purpose. Godio et al. [24] used SEIR models for predicting epidemic evolution by means of a stochastic solver, which allows a time-dependent transmission rate. They model the transmission rate as a function of community mobility. This approach is more flexible than the constant transmission rate assumption. However, it still cannot capture other dynamic aspects of the environment that impact the transmission rate, such as masks mandates, and adoption of contact tracing, early testing and isolation. Alternatively, Friston et al. [25] proposed a dynamic causal model framework for COVID-19, where they tried to include every variable that "matters" in the spread of the disease. This model suggested that individuals had four different characteristics: location, infection status, testing results, and clinical status (i.e., how sick they are). Each of these four characteristics contained four different states, and individuals could move from one state to another state over time. The main challenge was that there were many parameters used in the model, and identifying accurate initial estimates of all the parameters is difficult for a novel infectious disease with non-specific symptoms and potentially many asymptomatic cases.

The objective of this paper is to provide a method that can be reliably used to make predictions for the epidemic evolution in the next two to three weeks, based on the observed incidence cases only. Due to the relative small percentage of death in the whole population, we will ignore the death data in our modeling. The motivation for this work originated from pragmatic planning questions posed by local and state officials charged with allocating resources and ensuring population health. Members from the Texas A&M University School of Public Health started to monitor and forecast COVID-19 cases at the beginning of the pandemic, and then used the projected cases to support predictions for hospitalization and related health resource utilization.

## Methods

Assuming that we have observed a time series of COVID-19 incidence cases up to a time t, our goal is to make predictions of incidence cases in the next two to three weeks. In an ideal scenario, all data sets would be calibrated to the time of infection (an admitted impossibility). However, publicly available data sets most often reflect the date of reporting, which may be the date of reporting to the local health department, but more often reflects the date of reporting up the chain, such as to the State health department. As such, day-to-day variations of reported incidence cases often reflect not the true variation of the disease infection but reporting capacity. In addition, a large data dump might occur because of attempts to process backlogged data. Therefore, we propose to perform a smoothing average of data (e.g. 3-day weighted average) before performing any analysis. In the event of a big data dump, we also need to make adjustment to the data and distribute the cases over time. These adjustment to public databases would not only improve model handling but also be valuable for our interpretation and application.

Our approach to forecasting future COVID-19 cases involves two main steps. First, we model the observed incidence cases using similar ideas as appeared in Cori et al. [26]. Assuming a Poisson distribution for the daily incidence number, and a gamma distribution for the series interval, we are able to estimate the parameter (i.e. the effective reproduction number $R_e$) in the model. In the forecasting step, we draw future incidence cases from their posterior predictive distributions, assuming that the current $R_e$ will stay the same, decrease 5%, or increase 5%. The upper 95% posterior credible intervals for increased $R_e$ scenario together with the lower 95% posterior credible intervals (CI) for decreased $R_e$ scenario constitute our prediction intervals. The detailed description of our methods can be found in S1 Appendix.

Some basic assumptions are necessary for using our methods. In order to determine the value of the effective reproduction number $R_e$, we made the assumption that $R_e$ has a prior gamma distribution with a shape parameter of 1 and a scale parameter of 5, similar to Cori et al. [26]. We also assumed that the serial interval has a discretized gamma distribution [26] with a mean of 3·95 and a standard deviation of 4·24 [27]. These hyper-parameters are generally fixed in our model and in our projection.

One parameter that we allow to vary is the time interval $\tau$ which we use to get reliable estimates of $R_e$. In essence, we assume that $R_e$ is constant during this interval $[t - \tau + 1, t]$ so that we can get a reliable estimate of $R_e(t)$ at time $t$. From our experience, $\tau = 7$ days or $\tau = 12$ days are recommended, the choice of which depends on the incidence numbers (smaller incidence cases require a larger $\tau$) and the actual dynamic change of the transmission rate (a smaller $\tau$ can capture the change better). A detailed discussion of the assumptions and parameters used for our model is provided in the "Choosing Model Parameter" section in S1 Appendix.

## Application to COVID-19 data sets

We first demonstrate how to use our methods for predicting COVID-19 cases in Texas, a large and diverse state in the US with a population size of approximately 29 million. We utilize data from the COVID-19 Data Repository by the Center for Systems Science and Engineering (CSSE) at Johns Hopkins University. As of November 15, 2020, the total number of reported cases was 1,059,753, corresponding to an attack rate of 38·0 per 1,000 people.

We emphasize the importance of understanding how the case reports can be influenced by administrative issues, and the need to adjust our model accordingly. For example, on September 21, 2020 there was a reported 14,129 cases for Harris county due to processing of backlogged data on that day. This artificial spike would influence the estimate of $R_e$, and consequently, the prediction going forward. Therefore, we reassigned those cases from Harris county according to the following rule: We first imputed the number of cases on that day using the average number of cases in the past seven days. Then we evenly spread the extra cases over the previous 31 days including that index day of September 21. The modified series would be treated as the observed series in our subsequent modeling analysis. Another modification we made was to smooth the data series. Due to the high variability of the daily cases, and the fact that there was often a delay in reporting especially during the weekends, we smoothed the data using the following algorithm, similar to Sun et al. [28]:

$$I(t) = 0·3 * I(t-1) + 0·4 * I(t) + 0·3 * I(t+1), \quad t = 2, 3, \cdots, T-1$$

$$I(1) = 0·7 * I(1) + 0·3 * I(2),$$

$$I(T) = 0·7 * I(T) + 0·3 * I(T-1),$$

## Daily Incidence of COVID-19 and Estimated $R_e$ for Texas (7-day intervals)

**Fig 1. Texas incidence cases over time (smoothed) and the estimated effective reproduction number $R_e(t)$ (95% CI in shaded area) using 7-day intervals.**

where $T$ is the last time point in the data series upon which a forecast is to be made. The smoothed data series were the data we used for generating our prediction models.

As mentioned in the detailed "Methods" section in S1 Appendix, we first used the method of Cori et al. [26] to estimate the reproduction number $R_e(t)$ for different time $t$ based on the smoothed incidence data in Texas, with a cut off date of November 15, 2020, and an interval of $\tau = 7$ days. (Results using $\tau = 12$ days are presented in S1–S3 Figs). The smoothed data series, the estimated $R_e(t)$, and its 95% confidence intervals (CI) are shown in Fig 1.

It is clear from Fig 1 that there were different stages of COVID-19 spread in Texas. Due to the large number of incidence cases, the 95% CI for the effective reproduction number $R_e$ are quite narrow. During the month of April, the case counts were kept very low due to a statewide Shelter-in-Place order that was enacted by the Governor. The estimated $R_e$ was close to 1·0 around mid-April. Beginning May 1, 2020 Texas started phased reopening process, with many restrictions lifted in early June, right after the Memorial Day holiday. The daily incidence cases began to increase dramatically after Memorial Day weekend, and continued throughout June, reaching a peak daily incidence of about 13,000 in early July. During this period, $R_e$ gradually increased to a value of 1·325. A statewide mask mandate was implemented on July 3, 2020, and a couple of weeks after that, we started to see a downward trend in the incidence cases. The

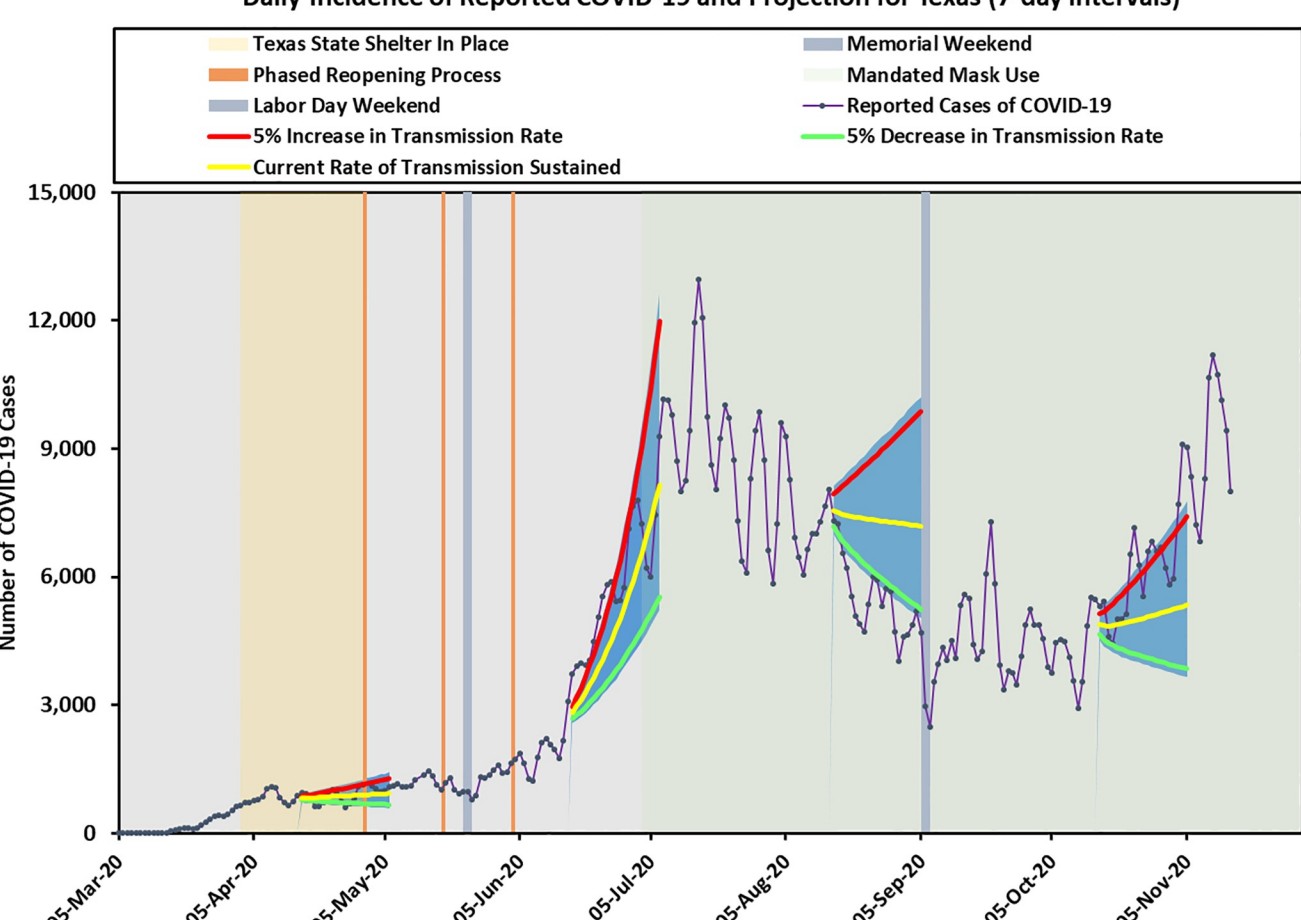

**Fig 2. Texas predicted incidence cases using 7-day intervals.** Three solid lines represent the predicted cases corresponding to current rate of transmission sustained, 5% increase in transmission rate, and 5% decrease in transmission rate. The shaded areas indicate prediction intervals.

reproduction number slowly decreased to below 1·0 towards the end of July and during August. Unfortunately, the trend reversed starting in early September, with cases increasing again and a reproduction number above 1.0. The uptick was possibly due to Labor Day weekend gatherings and widespread reopening of in-person options for schools and colleges for the Fall 2020 semester. The epidemic was then kept under control for a while until Mid-October, when COVID-19 cases started to increase dramatically both statewide and nationwide.

For illustration purposes, we applied our prediction method at four equally spaced time points that were two months apart: April 15, June 15, August 15, and October 15. We plotted three projection lines corresponding to the predicted mean values when the transmission rate (or equivalently the reproduction number $R_e$) stayed the same, increased 5%, or decreased 5%. We also plotted the prediction intervals (shaded areas) based on the upper 95% CI limits for the 5% increasing $R_e$ and the lower 95% CI limits for the 5% decreasing $R_e$ scenario. The predicted daily cases and cumulative cases, together with their prediction intervals for the next three weeks are shown in Figs 2 and 3 separately.

As expected, our predictions performed differently at different times. On April 15, our forecast assuming constant transmission rate matched the observed data very well. On June 15, when $R_e$ was increasing rapidly because of the business reopening process and the Memorial

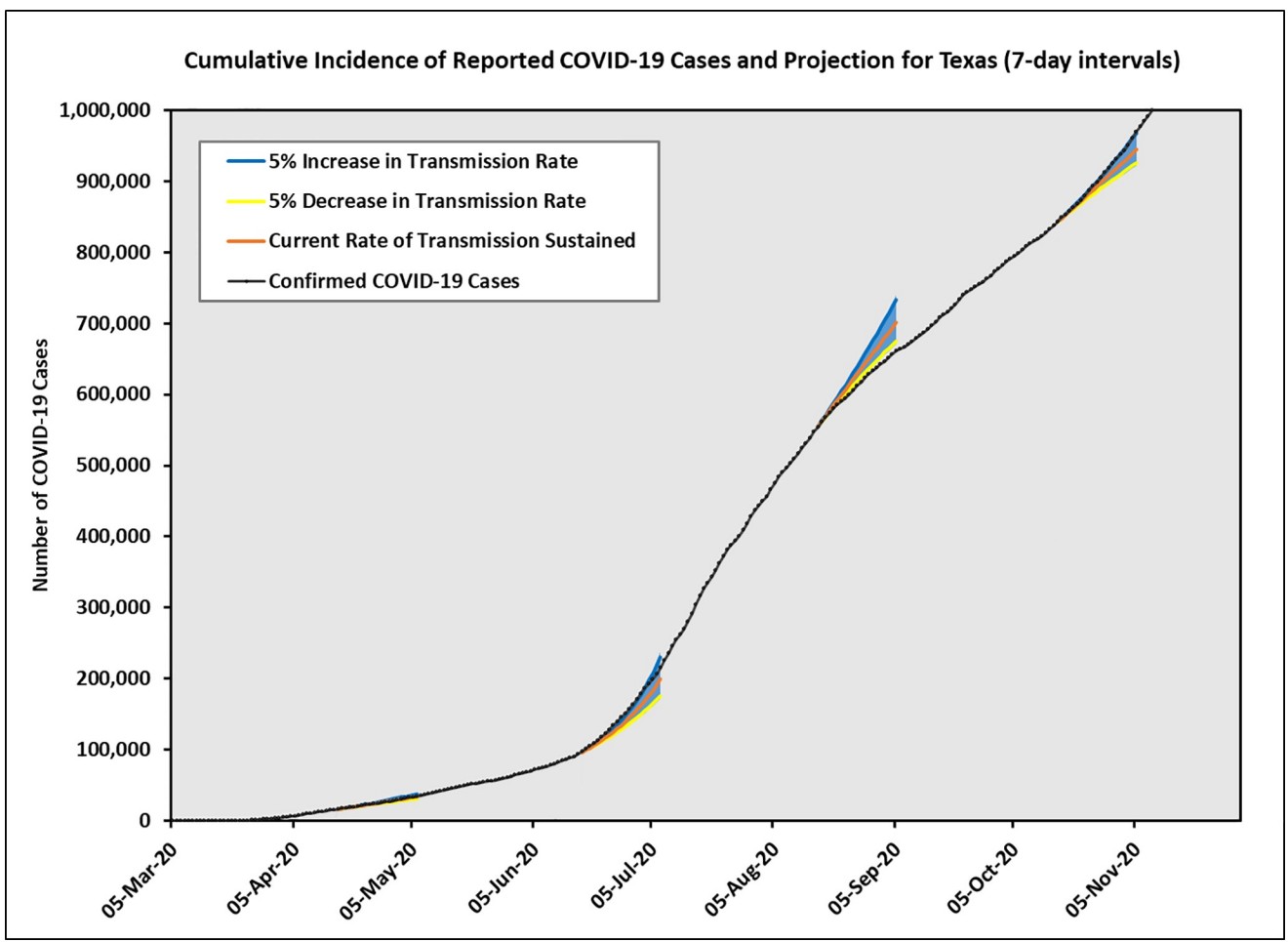

**Fig 3. Texas predicted cumulative incidence cases using 7-day intervals.** Three solid lines represent the predicted cases corresponding to current rate of transmission sustained, 5% increase in transmission rate, and 5% decrease in transmission rate. The shaded areas indicate prediction intervals.

Day holiday weekend, the observed cases fell between our predicted curves assuming the same transmission rate and 5% increase in transmission rate. On August 15, we saw a gradual decrease in transmission rate due to a statewide mask mandate, and the forecast with 5% decrease in transmission rate matched the observed data closely. Finally, on October 15, we started to see an increasing trend again, and the forecast assuming 5% increase in transmission rate worked well.

Secondarily, we chose to test the applicability of our model to a smaller geographic region within Texas. We applied our method to predicting the number of cases for the Brazos Valley (BV), a group of seven counties in Texas (i.e., Brazos, Robertson, Burleson, Madison, Grimes, Leon, and Washington counties), which collectively comprise the Bryan-College Station metropolitan area and neighboring counties. The center is Brazos County, where Texas A&M University is located. This area is approximately 100 miles from both Austin and Houston and has a younger population than Texas as a whole. Several healthcare entities and a public health authority in the BV needed timely and accurate forecasts to support planning for local COVID-19 cases.

The BV incidence cases and the estimated reproduction number $R_e(t)$ using 12-day intervals are presented in Fig 4. Due to small incidence cases in BV, the CIs for $R_e$ were quite wide,

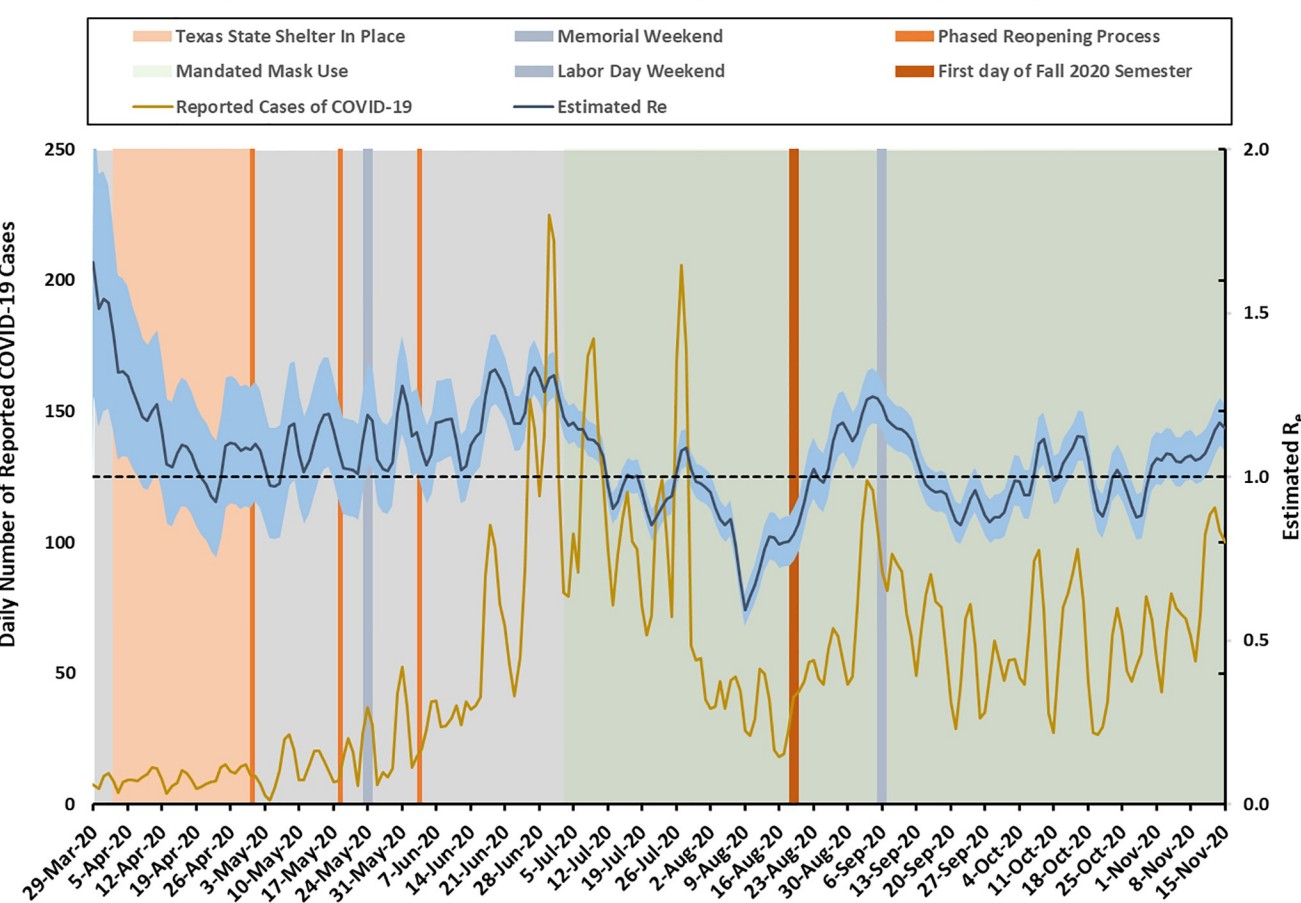

**Fig 4. Brazos Valley incidence cases over time (smoothed) and the estimated effective reproduction number $R_e(t)$ (95% CI in shaded area) using 12-day intervals.**

making forecasting for BV more challenging. The trend for BV was influenced by the local context so it did not always follow the trend in Texas. In addition, due to a relative small population size (approximately 229,000), and sudden population change caused by college students' moving out (in late-March corresponding to the Stay-at-Home order) and then back to the region (in mid-August to correspond with the start of the Fall semester), we saw more variability in the incidence cases for BV. Therefore, we chose to use 12-day intervals for our modeling approach, but we also provided results using 7-day intervals in S4–S6 Figs for additional information. All other parameters were the same as appeared in the state model, and we made predictions on the same days as we did for the state model. The predicted daily incidence cases and cumulative incidence cases for BV are shown in Figs 5 and 6 separately.

On April 15, our prediction assuming the same transmission rate sustained agreed well with the observed cases. On June 15, when the transmission rate increased rapidly, the prediction upper bounds followed approximately the observed curve. Our forecast based on past history did not capture the increased case numbers at the end of August when school started, since we had an influx of cases due to thousands of students moving to Brazos county from all over Texas. Starting October 15, although past trend suggested increasing incidence cases, the observed data matched more closely with the prediction lower bounds. Our model and

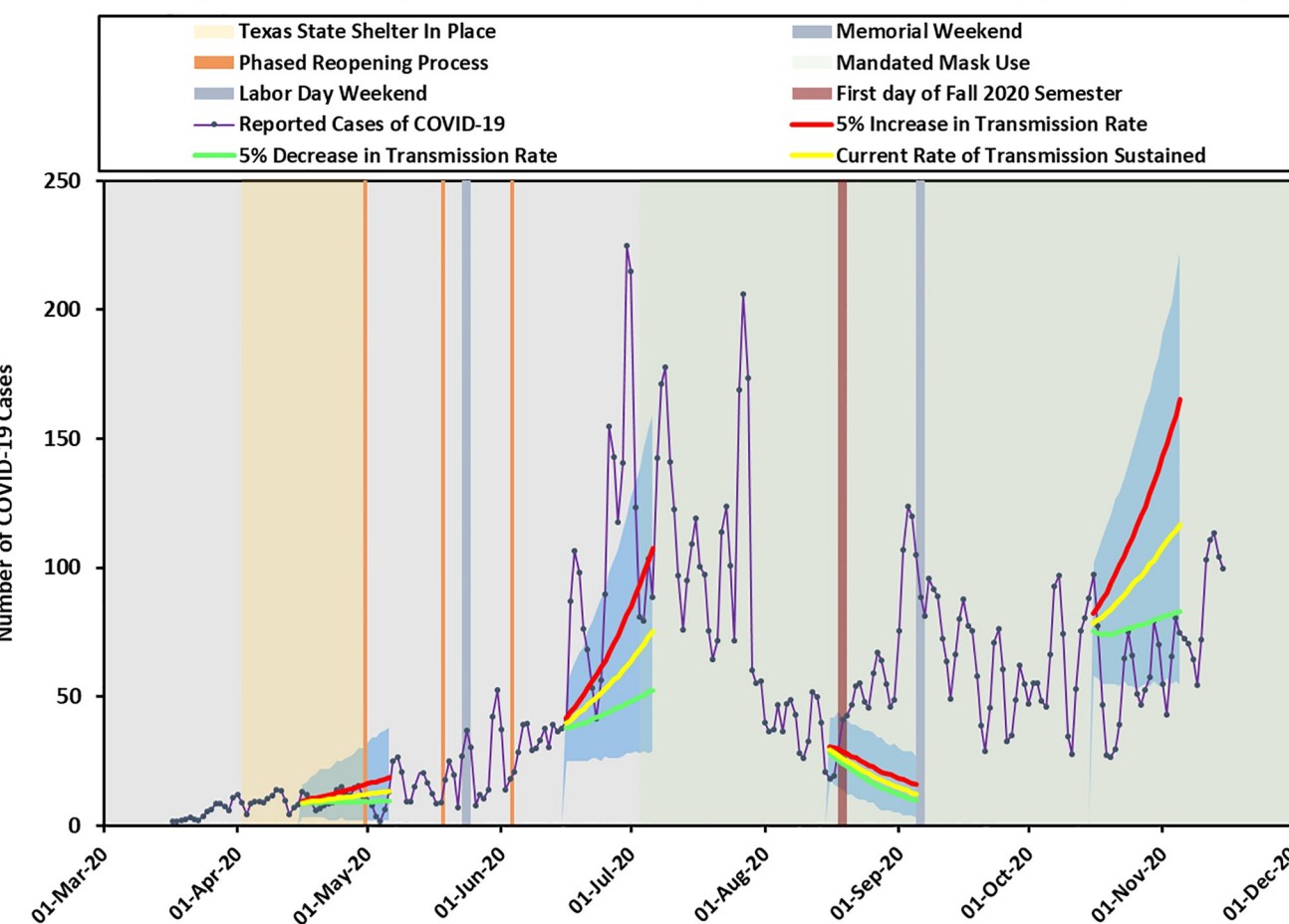

**Fig 5. Brazos Valley predicted incidence cases using 12-day intervals.** Three solid lines represent the predicted cases corresponding to current rate of transmission sustained, 5% increase in transmission rate, and 5% decrease in transmission rate. The shaded areas indicate prediction intervals.

method produced reasonably accurate results when the $R_e$ value is distributed similarly in the future as it is in the past. Large deviations from the predicted results can imply that a change in policy or some other factors have occurred that have dramatically altered the $R_e$ value over time.

## Conclusion

We have proposed a method that generates predictions for the number of COVID-19 infectious disease cases in the future, based on what estimates of $R_e$ are like at the current time. The major strength of our approach lies in its simplicity, which makes it easy to implement with a small team of modellers. As such, we have incorporated it as part of a dashboard (https://covid19-modeltrac.shinyapps.io/TX-BV-ModelTrac/#section-tx-forecasts), where it can automatically generate forecasting values every day for a future view of three weeks using publicly-available data. This transparent and straightforward approach means that the method can be easily adopted by others who want to do similar predictions to help inform local or state-wide decision using public data sources. Our predicted case numbers can also be used as data inputs alongside other information for predicting health care utilization and

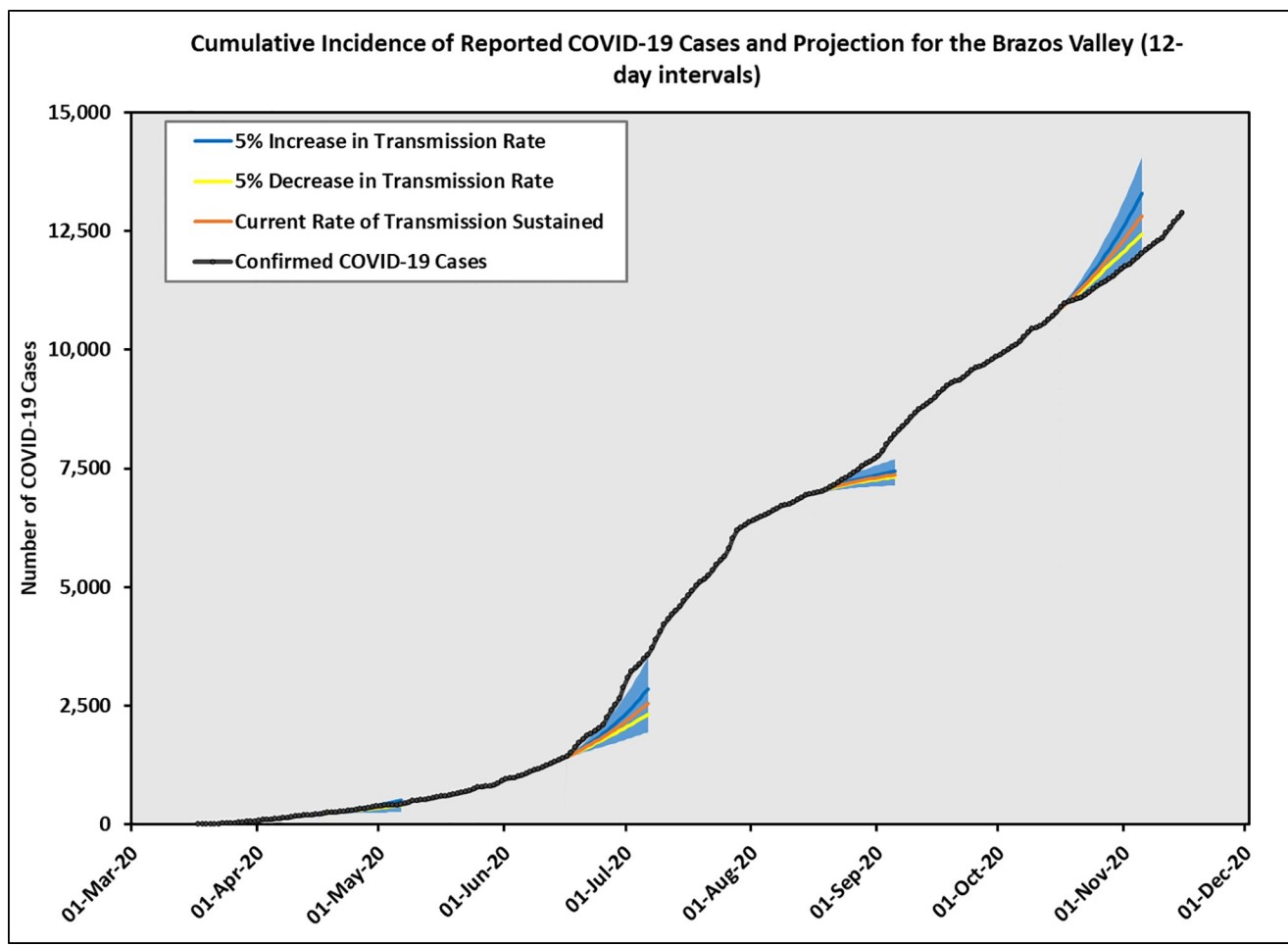

**Fig 6. Brazos Valley predicted cumulative incidence cases using 12-day intervals.** Three solid lines represent the predicted cases corresponding to current rate of transmission sustained, 5% increase in transmission rate, and 5% decrease in transmission rate. The shaded areas indicate prediction intervals.

health outcomes such as hospitalizations, intensive care unit (ICU) occupancy and corresponding ventilator use, and anticipated fatalities. These projections should be performed routinely to plan for surges and avoid overwhelming health resources. In Texas for example, hospitals are collectively working together using surge projections to identify and refer patients to available hospital beds [29].

A limitation for any infectious disease prediction model is the complexity inherent in how data are collected. Infectious disease reporting has long been plagued with many challenges. It is important to acknowledge that our model, as many others, relies on detection of infections through testing and reporting. In reality, the journey of a simple data element, from infection to tabulation, has many obstacles and nuances along the way. Some major complexities of the data include: policies about testing algorithms (e.g. which suspect cases are tested); if screenings or surveillance is conducted, which diagnostic test is acceptable or required for reporting; accessibility and availability of testing; administrative issues such as reporting requirements, procedures, and infrastructure. These elements can vary widely by locale and among populations within a locale. Thus, the available data are likely to represent some fraction of infections. Understanding the underlying caveats and how local situations

contribute to limitations is essential to evaluating the model output. Even so, the opportunity for practical application of our model to provide insight for assessment, planning, and policy-making remains invaluable.

Similar to the widely-adopted method for estimating $R_e$ [26], we made a few assumptions, e.g. the incidence $I(t)$ follows a Poisson distribution, with a mean parameter determined by a renewal function involving a serial function $w(s)$. The serial function is assumed to have a discretized gamma distribution. The reproduction number $R_e$ varies with time, but we assume that it is constant over a time interval (7 days, or 12 days) in order to obtain a stable estimate for its posterior distribution. Under these assumptions, we can predict the number of cases that could occur in the following two or three weeks, allowing $R_e$ to stay the same, increase 5%, or decrease 5%. The assumption that $R_e$ behaves similarly in the future as it does now is a major assumption, and is probably inaccurate if we project far into the future. However, we believe it to be a reasonable approximation of the true process if we want to see what happens in the next couple of weeks from the present.

Because $R_e$ is related to many factors, it can change dramatically. It is a function of transmission probability, which means it can be affected by a mask mandate. It is also affected by the average number of contacts one person has, hence, we expect that $R_e$ might increase when in-person school resumes. In addition, it depends on how many days on average one person is infectious after becoming infected, which can be reduced by contact tracing and early isolation. The number of people that are susceptible or immune is also changing over time. As more people become infected and then become recovered, the effective $R_e$ should decrease over time if other factors stay constant. If we want to make more accurate forecasts, we should allow a future $R_e$ to be a function of all these different factors. Another way to think about this is that if we make projections according to current values of $R_e$, then any deviations from the current trend can be attributed to factors not explicit in our model, such as a policy implementation, or behavior changes arising from reactions to current situation.

One contributing factor to $R_e$ that can be objectively measured is mobility data. If mobility data could provide insight on how $R_e$ may vary, incorporating the motility data in a prediction model can result in better predictions for $R_e$ in the future, which in turn will result in better estimates for the number of incidence cases. Finding the trend of $R_e$ values in the future using other data sources is a direction of our future research.

In summary, we presented a modelling approach that we believe can be easily adopted by others, and immediately useful for local or state planning. Although many initially downplayed the long-term consequences of COVID-19 [30], it is now clear that new surges are appearing in the US as well as globally [31–33], and that the pandemic spread is likely to last for another year or two [3]. Thus, public health and governmental responses will need to be guided by data that pinpoint where, when, and among whom the new cases are occurring. This information can help guide public health messaging as well as the nature and degree of government responses to mandating public health practices or regulating business operations to limit spread. Timely projections regarding case counts are critical to planning for healthcare resources and assuring available care and best possible outcomes for populations facing the uncertainty of a rapidly emerging infectious disease during a pandemic response.

## Supporting information

**S1 Appendix. Technical details.** Methods for predicting COVID-19 cases and the selection of model parameters.
(PDF)

**S1 Fig. Texas incidence cases over time (smoothed) and the estimated effective reproduction number $R_e(t)$ (95% CI in shaded area) using 12-day intervals.**
(TIF)

**S2 Fig. Texas predicted incidence cases using 12-day intervals.** Three solid lines represent the predicted cases corresponding to current rate of transmission sustained, 5% increase in transmission rate, and 5% decrease in transmission rate. The shaded areas indicate prediction intervals.
(TIF)

**S3 Fig. Texas predicted cumulative incidence cases using 12-day intervals.** Three solid lines represent the predicted cases corresponding to current rate of transmission sustained, 5% increase in transmission rate, and 5% decrease in transmission rate. The shaded areas indicate prediction intervals.
(TIF)

**S4 Fig. Brazos Valley incidence cases over time (smoothed) and the estimated effective reproduction number $R_e(t)$ (95% CI in shaded area) using 7-day intervals.**
(TIF)

**S5 Fig. Brazos Valley predicted incidence cases using 7-day intervals.** Three solid lines represent the predicted cases corresponding to current rate of transmission sustained, 5% increase in transmission rate, and 5% decrease in transmission rate. The shaded areas indicate prediction intervals.
(TIF)

**S6 Fig. Brazos Valley predicted cumulative incidence cases using 7-day intervals.** Three solid lines represent the predicted cases corresponding to current rate of transmission sustained, 5% increase in transmission rate, and 5% decrease in transmission rate. The shaded areas indicate prediction intervals.
(TIF)

## Acknowledgments

We are appreciative of the inspiration and insight we have gotten from the Texas A&M Emergency Management Advisory Group, and Public Health Modelling Team.

## Author Contributions

**Conceptualization:** Hongwei Zhao, Tiffany A. Radcliff, Murray J. Cote, Rebecca S. B. Fischer, Marcia G. Ory.

**Formal analysis:** Hongwei Zhao, Naveed N. Merchant.

**Funding acquisition:** Marcia G. Ory.

**Investigation:** Alyssa McNulty.

**Methodology:** Hongwei Zhao, Naveed N. Merchant.

**Project administration:** Marcia G. Ory.

**Software:** Hongwei Zhao, Naveed N. Merchant, Huiyan Sang.

**Supervision:** Marcia G. Ory.

**Visualization:** Murray J. Cote.

**Writing – original draft:** Hongwei Zhao, Naveed N. Merchant.

**Writing – review & editing:** Hongwei Zhao, Alyssa McNulty, Tiffany A. Radcliff, Murray J. Cote, Rebecca S. B. Fischer, Huiyan Sang, Marcia G. Ory.

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
