## [Decision Letter · Decision Letter 0]

28 Jan 2021

PONE-D-20-36327

COVID-19: Short term prediction model using daily incidence data

PLOS ONE

Dear Dr. Zhao,

Thank you for submitting your manuscript to PLOS ONE. After careful consideration, we feel that it has merit but does not fully meet PLOS ONE’s publication criteria as it currently stands. Therefore, we invite you to submit a revised version of the manuscript that addresses the points raised during the review process.

We look forward to receiving your revised manuscript.

Kind regards,

John Schieffelin, MD

Academic Editor

PLOS ONE

Journal Requirements:

2.Thank you for stating the following in the Acknowledgments Section of your manuscript:

"We thank Texas A&M University administration for internal funding to support this 264

work."

Reviewers' comments:

Reviewer's Responses to Questions

**Comments to the Author**

1. Is the manuscript technically sound, and do the data support the conclusions?

Reviewer #1: Partly

Reviewer #2: Yes

2. Has the statistical analysis been performed appropriately and rigorously? 

Reviewer #1: N/A

Reviewer #2: Yes

3. Have the authors made all data underlying the findings in their manuscript fully available?

Reviewer #1: Yes

Reviewer #2: Yes

4. Is the manuscript presented in an intelligible fashion and written in standard English?

Reviewer #1: Yes

Reviewer #2: Yes

5. Review Comments to the Author

Reviewer #1: The need for an accessible method to estimate and predict SARS-CoV2 incidence, both short- and long-term, is very real, and the authors propose an intriguing option to meet that need. Howerver, they themselves point out that the porposed model is less successful when a variety of parameters shift within the projection interval. Under these circumstances the range encompassed by the +/- 5% change seems, in fact, unacceptabley wide from an operational perspective, and not resonable as suggested by the authors. And while the authors identified a number of chages of status which could be resonble easy to identfiy (and avoid) for a predeiction period, they never mentioned one of the truly problematic elements related to identificaiton of SARS-CoV2 infections (cases), which is the testing itself - not just the mentioned lag in reproting, but actual uptake of testing, and the tremendous variablility that can occur in uptake of diagnostic testing, influenced by supply shortages, population interest in and access to testing, at a local or state level. The unfortunate reality is that diganosed and reported infections with SARS-CoV2 are in fact, some unknown fraction of true infections, which also changes over time. This model actual gives some evidence of that, providing much tighter ranges, aligning more closely with actual case counts in the early periods, with far less precision in the late intervals.

Reviewer #2: In the paper PONE-D-20-36327 "Covid-19, Short term prediction model using daily incidence data", Zhao et al proposed a new approach to forecasts the number of incident cases in the near future using some assumptions. Based on the paper, they reported that the method can produces reasonably results and large deviation from the predicted results can imply that a change in policy or some other factors. The results seem reasonable.

Some similar results have been studied by Jin's group in Fudan(See [CCJL2020],[SZYPCC2020],[P2020]), Jin's model is well suitable for Chinese data. But the scene and data in USA are more complicated. Zhao's work is interesting.

One suggestion is that we may not deal with the original number of incident cases, instead, we may consider to filter or smooth the number of incident cases, for example, 7-day average.

[CCJL2020]Chen, Y., Cheng, J., Jiang, Y. and Liu, K. A time delay

dynamical model for outbreak of 2019-nCoV and the parameter

identification. J. Inverse Ill-Posed Probl., 28(2020), 243–250.

[SZYPCC2020]Shao, N., Zhong, M., Yan, Y., Pan, H., Cheng, J. and Chen, W.

Dynamic models for coronavirus disease 2019 and data

analysis. Math. Methods Appl. Sci., 43(2020), 4943–4949.

[P2020]Hanshuang Pan, Nian Shao, Yue Yan, Xinyue Luo, Shufen Wang, Ling Ye, Jin Cheng and Wenbin Chen,

Multi-chain Fudan-CCDC model for COVID-19-a revisit to Singapore's case,Quantitative Biology, 2020, 8(4): 325–335.

6. PLOS authors have the option to publish the peer review history of their article (what does this mean?). If published, this will include your full peer review and any attached files.

Reviewer #1: No

Reviewer #2: No

---

## [Author Response · Author response to Decision Letter 0]

11 Feb 2021

We thank the reviewers and the academic editor for valuable comments on our submitted manuscript entitled "COVID-19: Short term prediction model using daily incidence data" (PONE-D-20-36327). We have made changes according to your requests and replied to the questions in the file named "Response to Reviewers.docx". We also uploaded two copies of our manuscript, one was a marked-up copy that highlighted changes made to the original version, and an unmarked version without tracked changes. 

Thank you very much for your consideration. Look forward to hearing back from you soon.

---

## [Decision Letter · Decision Letter 1]

31 Mar 2021

COVID-19: Short term prediction model using daily incidence data

PONE-D-20-36327R1

Dear Dr. Zhao,

We’re pleased to inform you that your manuscript has been judged scientifically suitable for publication and will be formally accepted for publication once it meets all outstanding technical requirements.

Kind regards,

John Schieffelin, MD

Academic Editor

PLOS ONE

Additional Editor Comments (optional):

Reviewers' comments:

Reviewer's Responses to Questions

**Comments to the Author**

1. If the authors have adequately addressed your comments raised in a previous round of review and you feel that this manuscript is now acceptable for publication, you may indicate that here to bypass the “Comments to the Author” section, enter your conflict of interest statement in the “Confidential to Editor” section, and submit your "Accept" recommendation.

Reviewer #1: All comments have been addressed

Reviewer #2: All comments have been addressed

2. Is the manuscript technically sound, and do the data support the conclusions?

Reviewer #1: (No Response)

Reviewer #2: Yes

3. Has the statistical analysis been performed appropriately and rigorously? 

Reviewer #1: (No Response)

Reviewer #2: Yes

4. Have the authors made all data underlying the findings in their manuscript fully available?

Reviewer #1: (No Response)

Reviewer #2: Yes

5. Is the manuscript presented in an intelligible fashion and written in standard English?

Reviewer #1: (No Response)

Reviewer #2: Yes

6. Review Comments to the Author

Reviewer #1: (No Response)

Reviewer #2: In the paper "COVID-19: Short term prediction model using daily incidence data", they describe a new approach that forecasts the number of incident cases, first model the observed incidence cases using a Poisson distribution for the daily incidence number, and a gamma distribution for the series interval, then estimate the effective reproduction number assuming its value stays constant during a short time interval; and finally draw future incidence cases from their posterior distributions.

The method is interesting and new, and the forecast results and explanation seem reasonable.

7. PLOS authors have the option to publish the peer review history of their article (what does this mean?). If published, this will include your full peer review and any attached files.

Reviewer #1: No

Reviewer #2: No

---

## [Editor Report · Acceptance letter]

5 Apr 2021

PONE-D-20-36327R1 

COVID-19: Short term prediction model using daily incidence data 

Dear Dr. Zhao:

I'm pleased to inform you that your manuscript has been deemed suitable for publication in PLOS ONE. Congratulations! Your manuscript is now with our production department. 

Kind regards, 

on behalf of

Dr, John Schieffelin 

Academic Editor

PLOS ONE